# Lithium Prevents Telomere Shortening in Cortical Neurons in Amyloid-Beta Induced Toxicity

**Rafael M. Themoteo** [1,*], **Vanessa J. R. De Paula** [1,2,*], **Nicole K. R. Rocha** [1,2], **Helena Brentani** [2] 
and **Orestes V. Forlenza** [1]

1 Laboratory of Neuroscience (LIM-27), Departamento e Instituto de Psiquiatria, Hospital das Clinicas HCFMUSP, Faculdade de Medicina, Universidade de São Paulo, Sao Paulo 05508-090, SP, Brazil

2 Laboratory of Psychobiology (LIM-23), Instituto de Psiquiatria, Hospital das Clinicas HCFMUSP, Faculdade de Medicina, Universidade de São Paulo, Sao Paulo 05508-090, SP, Brazil

* Correspondence: rafael.themoteo@fm.usp.br (R.M.T.); vanessaj@usp.br (V.J.R.D.P.); Tel.: +55-(11)2661-6978 (V.J.R.D.P.)

**Abstract:** Background: There is consistent evidence of the potential benefits of lithium attenuating mechanisms of neurodegeneration, including those related to the pathophysiology of Alzheimer's disease (AD), and facilitating neurotrophic and protective responses, including maintenance of telomere length. The aim was to investigate the protective effect of the pre-treatment with lithium on amyloid-beta (Aβ)-induced toxicity and telomere length in neurons. Methods: Cortical neurons were treated with lithium chloride at therapeutic and subtherapeutic concentrations (2 mM, 0.2 mM and 0.02 mM) for seven days. Amyloid toxicity was induced 24 h before the end of lithium treatment. Results: Lithium resulted in 120% (2 mM), 180% (0.2 mM) and 140% (0.02 mM) increments in telomere length as compared to untreated controls. Incubation with $A\beta_{1-42}$ was associated with significant reductions in MTT uptake (33%) and telomere length (83%) as compared to controls. Conclusions: Lithium prevented loss of culture viability and telomere shortening in neuronal cultures challenged with Aβ fibrils.

**Keywords:** telomere; lithium; cortical neurons; amyloid-beta; Alzheimer's disease

## 1. Introduction

Telomeres are terminal DNA regions that lose 50–150 pb fragments at each cycle of cell division. As a consequence, telomeres work as a 'biological clock', avoiding the endless proliferation of cells and establishing replicative senescence [1].Telomere shortening may also occur as a consequence of oxidative stress in the brain, along with protein oxidation and lipid peroxidation [2,3]. Evidence of telomere shortening has been reported in age-associated diseases, including Alzheimer's (AD) and related neurodegenerative disorders [4].

Pharmacological compounds that hinder or attenuate telomere shortening by oxidative stress and other mechanisms, therefore preserving cell function and survival of neuronal cells, may be helpful in the treatment or prevention of age-related neurodegenerative disorders. There is a substantial body of evidence from experimental and clinical models suggesting that lithium has additional neurotrophic and protective properties [5,6]; these properties may be relevant to disease modification in certain neuropsychiatric and neurodegenerative diseases [7], and AD [6,8].

Therefore, the overall effect of lithium may have specific disease-modifying properties in AD by tackling core pathophysiologic mechanisms [8,9] and also deliver a myriad of unspecific yet potentially relevant clinical benefits through its multiple effects on cellular function, including the regulation of telomere homeostasis. In a recent study conducted in our group, using a triple-transgenic mouse model of AD, we were able to provide evidence

of a dose-dependent, the tissue-specific effect of long-term lithium treatment on the maintenance of telomeres in the presence of AD pathology [10]. We also show that lithium (1 and 2 mM) modifies ontological pathways in hippocampal tissue from triple transgenic animals for AD, directly related to neuroprotective responses. Among the modifications, we have neuronal projection pathways, regulation of glial cell activity and neuroinflammation [11]. Furthermore, acute lithium treatment induced an increase in Wnt/β-catenin pathway transcripts (HIG2, Bcl-xL, Cyclin D1 and c-myc) in the hippocampus and cerebral cortex of adult mice [12]. Low-dose of lithium significantly improve spatial memory and ameliorate cognitive dysfunction and pathological alterations in Alzheimer's disease transgenic mice (APP/PS1) [13] and stimulates hippocampal glucose metabolism [14]. In the present study, we address the protective effect of chronic lithium treatment on Aβ-induced telomere shortening in cultured cortical neurons.

## 2. Materials and Methods

### 2.1. Compliance with Ethical Standards

All animal experiments were approved by the local Ethics Committee (CAPPesq n°1293/09) in accordance with Directive 2018/63/EU, the committee of the University of Sao Paulo Medical School, Brazil, under protocol number n°1293/09. All national guidelines were taken into consideration.

### 2.2. Primary Cultures of Cortical Neurons

Primary neuronal cultures were prepared according to the [6,12,15,16] method. Treatment with beta-amyloid peptide (Aβ$_{1-42}$) was done for 24 h from the 9th to the 10th day in culture, as described elsewhere [17]. To assess the effect of lithium on telomere length in amyloid-challenged cultures, we used a working concentration of 5 μM of the Aβ$_{1-42}$ peptide, previously incubated to turn into the fibrillary state [17]. After ten days in culture, neuronal viability was microscopically ascertained prior to harvesting the cultured cells upon completion of the incubations.

### 2.3. Assessment of Cell Viability

The viability of neuronal cultures was quantitatively assessed by the MTT method [(3-(4,5-dimethylthiazol-2-yl)-2,5-diphenyltetrazolium bromide], which estimates the percentage of living cells in a given substrate compared to experimental control. Briefly, 50 μL of MTT solution (5 mg/mL in PBS) was added to each well (1 × 10$^5$ cells/mL), and multi-well plates were incubated for 3 h at 37 °C and 5% CO$_2$. Then, 500 μL of 10% sodium-dodecyl sulfate (SDS) in 0.01N HCl was added. After overnight incubation, the absorbance was measured by spectrophotometry at 570 nm.

### 2.4. DNA extraction

We used the AllPrepDNA/RNA mini kit (QIAGEN, Hilden, Germany) following the manufacturer's guidelines. The samples were quantified by spectrophotometry with the NanoDrop apparatus (ThermoFisher Scientific, Waltham, MA, USA), and the samples were diluted to reach the final concentration of 25 ng/μL of DNA.

### 2.5. Amplification primers

Two solutions were prepared, one for the amplification of the 36B4 gene (control gene) and another for that of the telomere region (region of interest). For a single amplification reaction, we used 8.18 μL of PowerUp SYBR Green Master Mix (ThermoFisher Scientific, Waltham, MA, USA), 0.44 μL of primer forward (Exxtend, Paulínia, Brazil) [5′ CGGTTTGTTTGGGTTTGGGTT TGGGTTTGGGTTTGGGTT 3′], 1.47 μL of primer reverse (Exxtend, Paulínia, Brazil) [5′ GGCTTGCCTTACCCTTACCCTTACCCTTACCCTTACCCT 3′] and 4.97 μL of RNAase-free water, in order to complete 15 μL. For a unique amplification reaction of 36B4, we use 8.18μL of PowerUp SYBR Green Master Mix (ThermoFisher Scientific, Waltham, MA, USA), 0.49 μL of primer forward (Exxtend, Paulínia, Brazil) [5′

CAGCAAGTGGGAAGGTGTAATCC 3′], 4.10 µL de primer reverse (Exxtend, Paulínia, Brazil) [5′ CCCATTCTATCATCAACGGGTACAA 3′] e 2.30 µL of RNAase-free water, in order to complete 15 µL. We adjusted the apparatus to a cycle of 2 min at 50 °C, 2 min at 95 °C, 15 s at 95 °C and 1 min at 60 °C and the other instructions were followed according to the protocol of the manufacturer.

### 2.6. Telomere and 36B4 Amplification

Telomere amplification reactions were performed using 7.5 µL of PowerUp SYBR Green Master Mix (ThermoFisher Scientific, Waltham, MA, USA), 0.45 µL of primer forward (Exxtend, Paulínia, Brazil) [5′-GGTTTTTGAGGG TGAGGGTGAGGGTGAGGGTGAGGGT-3′], 0.45 µL of primer reverse (Exxtend, Paulínia, Brazil) [5′–TCCCGACTATCCCTA TCCCTATCCCTATCCCTATCCCTA-3′] and 3 µL of water to complete 15 µL. For the 36B4 amplification reaction, we used 7.5 µL of PowerUp SYBR Green Master Mix (Thermo Fisher Scientific, Paulínia, Brazil), 0.45 µL de primer forward (Exxtend, Paulínia, Brazil) [5′CAGCAAGTGGGAAGGTGTAATCC 3′], 0.75 µL of primer reverse (Exxtend, Paulínia, Brazil) [5′CCCATTCTATCATCAACGGGTACAA 3′] and 3.3 µL of water to complete 15 µL. The 7500 Fast Real-Time PCR System (Thermo Fisher Scientific, Waltham, MA, USA) was set to a holding stage of 2 min at 50 °C, 2 min at 95 °C, 15 s at 95 °C and 1 min at 60 °C. Each of these reactions was amplified in duplicates, calibrated with the negative control (water only) for each of the genomic regions of interest, yielding 172 reactions to telomere assessment and the same number for the control gene, 36B4. 96-well plates were assembled for each of the regions of interest, following the same protocol: in each well, 15 µL of the corresponding amplifier solution and 2 µL of the sample were to be amplified or water in the case of negative controls.

### 2.7. Statistical Analysis

An independent sample Student's *t*-test was used to determine the difference between MTT mean values in treatment conditions compared to respective controls. The analysis of telomere size was done according to the 2-ΔΔCT method adapted, comparing (quadruplicates samples) the target region (telomeres) with the control region (36B4). For each of the quadruplicates of the samples, we did the mean amplification CT of telomere and 36B4. Then, we calculated the difference between these mean CTs, and we obtained the ΔSAMPLE_CT (CT_TEL_SAMPLE–CT_36B4_SAMPLE). Thereafter, we did the mean CT of each condition and for each target and the difference between them (TEL_MEAN_CT–36B4_MEAN_CT) or ΔMEAN_CT. Finally, we obtained the ΔΔCT, that is, the difference between ΔSAMPLE_CT and ΔMEAN_CT, and we calculated the potency 2-ΔΔCT. The graphical analysis was done by the median potency 2-ΔΔCT for each of the conditions.

## 3. Results

Lithium treatment was associated with increased viability of cultured cortical neurons, as indicated by MTT assays. This effect was dose-dependent, although statistical significance was only achieved with the middle dose of lithium chloride (0.2 mM), in which case there was a 34% increase in MTT uptake (Figure 1). Treatment with Aβ$_{1-42}$ (5 µM) significantly impaired neuronal viability. This effect was partially (0.02 mM, increased 32%) or totally (0.2 mM and 2 mM increased 57% and 65% respectively) reverted by the pre-treatment of cultures with lithium chloride for seven days (Figure 2), all such effects reaching statistical significance compared to the effect of Aβ$_{1-42}$ alone (Figure 3).

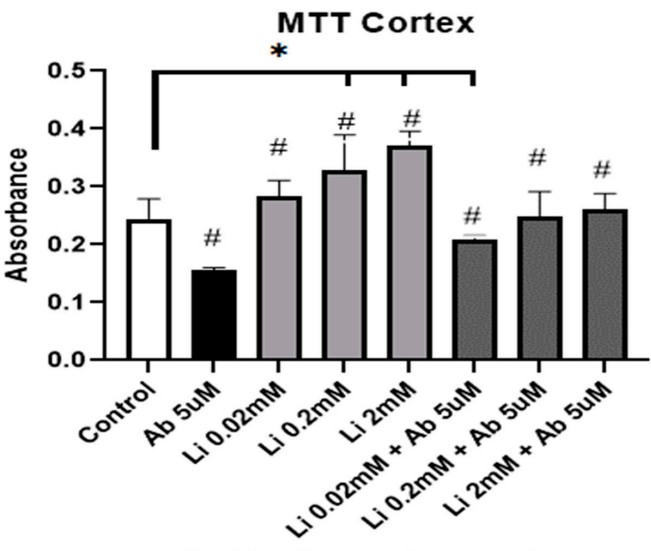

**Figure 1.** Graph representative of MTT [(3-(4,5-dimethylthiazol-2-yl)-2,5-diphenyltetrazolium bromide] absorbance in primary cultures of rat cortical neurons treated with different concentrations of lithium (0.02 mM; 0.2 mM and 2 mM) and neurons co-treated with lithium chloride and 5μM $A\beta_{1-42}$. The experimental controls are depicted in the leftmost (white) bar, indicative of untreated neurons, and the black bar represents the positive control for $A\beta_{1-42}$ toxicity. Light gray bars indicate the effect of lithium treatment alone on MTT uptake, as compared to untreated controls. Dark gray bars refer to the effect of lithium on neurons pre-treated with lithium and challenged with fibrillary $A\beta_{1-42}$ (independent samples *t*-tests * $p < 0.05$ vs. control, # $p = 0.05$ vs. $A\beta_{1-42}$). The raw data are described in Table A1.

**Telomere length - 2^ΔΔCT**

**Figure 2.** Estimates of telomere length in primary cultures of rat cortical neurons treated with different concentrations of lithium (0.02 mM, 0.2 mM and 2 mM) and neurons co-treated with lithium chloride and $A\beta_{1-42}$ (5 μM). The experimental controls are depicted by the leftmost (white) bar, indicative of untreated neurons, and the black bar represents the positive control for $A\beta_{1-42}$ toxicity (83% reduction in telomere length compared to untreated controls). Light gray bars indicate the effect of lithium treatment alone on telomere length as compared to untreated controls. Dark gray bars refer to the effect of lithium in neurons challenged with fibrillary $A\beta_{1-42}$, as compared to the positive control, showing 136% and 179% increase in telomere length in neurons pre-treated with 0.02 mM and 0.2 mM lithium, respectively. $\Delta\Delta CT$, # $p < 0.05$ vs. control, * $p = 0.05$ vs. $A\beta_{1-42}$. The raw data are described in Table A2.

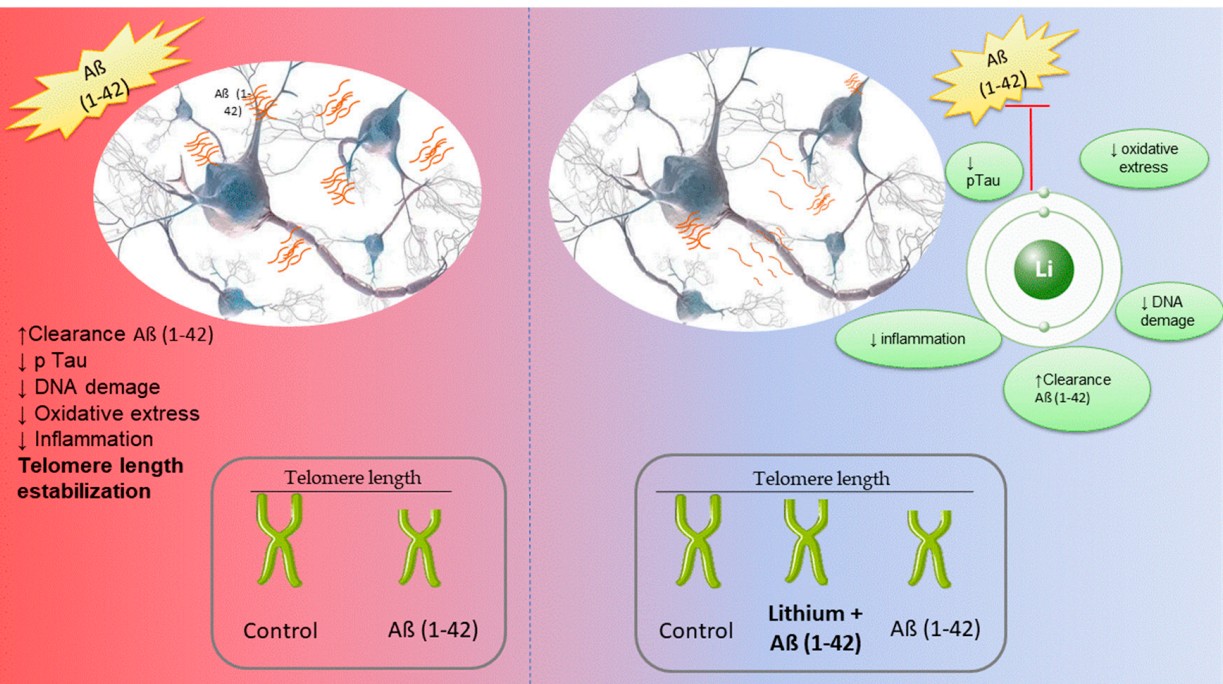

**Figure 3.** Representative image of the protective effect of lithium on cortical neurons, changes in cell viability and effect with telomere size.

## 4. Discussion

In the present study, we show that fibrillary amyloid-beta caused a statistically significant decrease in cell viability and telomere length in primary cortical neurons; these effects were partially or totally reverted by the pre-treatment of cultures with lithium chloride for seven days (chronic treatment). Our data reinforce the evidence supporting the multi-modal neuroprotective effects of lithium against amyloid toxicity, as reported in distinct experimental models [18], and further suggest that the overall neurotoxic effect of $A\beta_{1-42}$, as depicted by a decrease in culture viability, is accompanied by impairments in telomere homeostasis. In this regard, lithium prevented not only amyloid-induced telomere shortening but also was able to promote telomere elongation in spite of the presence of $A\beta_{1-42}$. This effect was observed at all working concentrations of lithium, i.e., ranging from micromolar to therapeutic levels.

Abnormal telomere shortening leads to premature senescence, cell arrest and loss of physiological functions [4]. Amyloid toxicity has been associated with telomere shortening mostly via mechanisms related to inflammation, oxidative stress and DNA degradation [19]. However, a study by [20] showed that intracellular $A\beta$ co-localizes in telomeres, inducing cell senescence and telomere shortening independently of the aforementioned mechanisms, supporting that the inhibition of telomerase activity may be an additional factor related to $A\beta$-induced cytotoxicity.

Telomerase, a key enzyme in the maintenance of telomere integrity, is regulated by its own catalytic subunit, the telomerase reverse transcriptase (TERT). Zhang et al. showed that the transcription of the *hTERT* gene is positively regulated by components of the Wnt/β-catenin pathway, leading to its increased expression. Whilst glycogen synthase-kinase 3-beta (GSK3β) phosphorylates β-catenin, leading to its degradation by the ubiquitin-proteasome pathway [21], its inhibition by lithium yields the retention of β-catenin and further downstream effects on the transcriptional activity of the *hTERT* gene [10]. Previous studies from our group and others indicate that lithium modulates multiple intracellular signaling cascades, with well-established effects such as inhibition of GSK3β [4]; activation of Wnt/beta-catenin signaling [12,21]; increase in the synthesis and release of neurotrophic factors, particularly brain-derived neurotrophic factor (BDNF) [15]; inhibition of apoptosis [21]. Lithium also increases metabolic efficiency and respiratory rate in mitochondria [22]

and modulates inflammatory response [6,23] and oxidative stress [24]. Therefore, multiple biological effects of lithium may converge to the modulation of telomere homeostasis, in addition to the up-regulation of other neurotrophic/protective responses.

Our data in primary cortical neurons suggest that chronic lithium treatment alone may lead to a substantial increase in telomere length (4- to 7-fold, dose-dependent increments), although this effect failed to reach statistical significance in view of the high variability of results. This effect would presumably reach statistical significance by increasing the number of replicates, which is unfortunately not available at the present time and represents a limitation of the study.

In summary, our study shows that pre-treatment with lithium protected neurons against telomere shortening induced by $A\beta_{1\text{-}42}$, restoring parameters similar to (or even higher than) baseline measures. Such effect was also observed with low working concentrations of lithium (0.02 mM and 0.2 mM), corroborating the notion that subtherapeutic or even lower (micromolar) concentrations of lithium may be effective in the modulation of biological responses [10,25]. In this regard, the present set of data suggests that the maintenance of telomere length is an additional mechanism by which lithium exerts neuroprotection, attenuating $A\beta$ toxicity, with potentially relevant implications for the treatment and prevention of cognitive decline and dementia in AD. Future directions: new experiments in neurons from humans with Alzheimer's disease will be used in ongoing studies. With this, we aim to explain how lithium protects the shortening of human telomeres.

**Author Contributions:** R.M.T.: Performed all cell cultures, data analysis and preparation of the manuscript. V.J.R.D.P.: Performed all cell cultures, data analysis and preparation of the manuscript. N.K.R.R.: Performed all cell viability tests, data analysis and preparation of the manuscript. H.B.: Data analysis and preparation of the manuscript. O.V.F.: Data analysis and preparation of the manuscript. All authors have read and agreed to the published version of the manuscript.

**Funding:** This work was supported by: Conselho Nacional de Desenvolvimento Científico e Tecnológico (CNPq), Grant No. 466625/2014-6; National Institute for Biomarker Research in Neuropsychiatry (INBION), Grants Nos. 14/50873-3, 2016/01302-9, 2017/14418-8, 2018/08621-8 (FAPESP), 88887.463672/2019-00 PNPD/CAPES (Programa Nacional de Pós-Doutorado/Capes) and 465412/2014-9 (CNPq); Associação Beneficente Alzira Denise Hertzog da Silva (ABADHS).

**Institutional Review Board Statement:** All animal experiments were approved by the local Ethics Committee (CAPPesq n°1293/09) in accordance with Directive 2018/63/EU, the committee of the University of Sao Paulo Medical School, Brazil, under protocol number n°1293/09. All national guidelines were taken into consideration.

**Informed Consent Statement:** Not applicable.

**Data Availability Statement:** The raw data for the realization of the average will be made available if requested by the authors.

**Conflicts of Interest:** All authors have no conflict of interest and are in accordance with the publication.

## Appendix A

**Table A1.** Absorbance in cortical cultures.

| Condition | Median Absorbance | % | Standard Deviation | Control *t* Test | Toxicity *t* Test 5 μM | Toxicity *t* Test 10 μM |
|---|---|---|---|---|---|---|
| Control | 0.245 | 1.000 | 0.024 | | | |
| aβ 5 μM | 0.156 | 0.636 | 0.003 | 0.001032 | | |
| Aβ 10 μM | 0.112 | 0.459 | 0.010 | 0.000035 | | |
| Li 0.02 mM | 0.284 | 1.158 | 0.020 | 0.063376 | 0.000046 | 0.000013 |
| Li 0.02 mM + aβ 5 μM | 0.209 | 0.852 | 0.006 | 0.041351 | 0.000007 | |
| Li 0.02 mM + aβ 10 μM | 0.172 | 0.704 | 0.020 | 0.006235 | | 0.001828 |
| Li 0.2 mM | 0.329 | 1.344 | 0.051 | 0.026209 | 0.000647 | 0.000216 |
| Li 0.2 mM + aβ 5 μM | 0.248 | 1.012 | 0.037 | 0.458576 | 0.002820 | |
| Li 0.2 mM + 10 aβ μM | 0.212 | 0.865 | 0.005 | 0.053733 | | 0.000004 |
| Li 2 mM | 0.370 | 1.513 | 0.019 | 0.000547 | 0.000002 | 0.000001 |
| Li 2 mM + aβ 5 μM | 0.260 | 1.062 | 0.021 | 0.258798 | 0.000173 | |
| Li 2 mM + ab 10 β μM | 0.236 | 0.965 | 0.035 | 0.383682 | | 0.000726 |

**Table A2.** TEL/36B4 ratio in Cortical.

| Condition | Median TEL/36B4 | % | Standard Deviation | Control *t* Test | Toxicity *t* Test |
|---|---|---|---|---|---|
| CTXCt | 1.21 | 100% | 0.058 | | |
| CTXaβ | 1.28 | 1.058 | 0.139 | 0.265 | |
| CTXLi2 | 1.05 | 0.871 | 0.092 | 0.026 | 0.035 |
| CTXLi0.2 | 1.06 | 0.878 | 0.071 | 0.021 | 0.037 |
| CTXLi0.02 | 1.08 | 0.895 | 0.055 | 0.019 | 0.041 |
| CTXaβLi2 | 1.13 | 0.939 | 0.061 | 0.034 | 0.053 |
| CTXaβLi0.2 | 1.11 | 0.919 | 0.053 | 0.014 | 0.035 |
| CTXaβLi0.02 | 1.12 | 0.927 | 0.042 | 0.019 | 0.046 |

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
