# Peer review of "Lithium Prevents Telomere Shortening in Cortical Neurons in Amyloid-Beta Induced Toxicity"

_neurosci, doi:10.3390/neurosci4010001_

Round 1
Reviewer 1 Report
In order to better analyse variations in cell viability with the different treatments (Aβ, Lithium, Lithium+ Aβ), authors should represent absorbance values (optical density) rather than percentages. In this way, reviewers (and readers) can figure out the degree of reproducibility of the data with the experiments performed (at least three).
Authors must explain the Statistical analyses they have performed with the telomere length data.
Author Response
Reviewer 1
In order to better analyse variations in cell viability with the different treatments (Aβ, Lithium, Lithium+ Aβ), authors should represent absorbance values (optical density) rather than percentages. In this way, reviewers (and readers) can figure out the degree of reproducibility of the data with the experiments performed (at least three).
Thanks for the suggestion, as suggested by reviewer 1 and 2, the graphs were changed in prism and the new version with the number of experiments (N=4) was added.
Authors must explain the Statistical analyses they have performed with the telomere length data.
We added a more detailed telomere length analysis on page 3.
“For each of the quadruplicates of the samples, we did the mean amplification CT of telomere and 36B4. Then, we calculated the difference between these mean CTs and we obtained the ΔSAMPLE_CT (CT_TEL_SAMPLE – CT_36B4_SAMPLE). Thereafter, we did the mean CT of each condition and for each target and the difference between them (TEL_MEAN_CT – 36B4_MEAN_CT), or ΔMEAN_CT. Finally, we obtained the ΔΔCT, that is, the difference between ΔSAMPLE_CT and ΔMEAN_CT, and we calculate the potency 2-ΔΔCT. The graphical analysis was done by the median potency 2-ΔΔCT for each of the conditions.”
Reviewer 2 Report
The manuscript presented from Rafael M. Themoteo et al., entitled "Lithium prevents telomere shortening in cortical neurons in amyloid-beta induced toxicity" is interesting. However there are some critical points that the authors should improve: 1) The introduction should be improved, the authors should mention previous studies that used lithium in neurodegenerative animal model 2) The authors should improve the presentation of their data, I would suggest to use a program as Graphpad or R to report the graph (and they should insert a table with raw data of qPCR) 3) If possible the authors should insert an image of cell (neuron) in culture at several concentration of lithiumAuthor Response
The manuscript presented from Rafael M. Themoteo et al., entitled "Lithium prevents telomere shortening in cortical neurons in amyloid-beta induced toxicity" is interesting. However there are some critical points that the authors should improve:
- The introduction should be improved, the authors should mention previous studies that used lithium in neurodegenerative animal model
Thanks for the suggestion, we added the following text on page 2:
“We also show that lithium (1 and 2 mM) modifies ontological pathways in hippocampal tissue from triple transgenic animals for AD, directly related to neuroprotective responses. Among the modifications, we have: neuronal projection pathways, regulation of glial cell activity and neuroinflammation (Rocha et al., 2020). Furthermore, acute lithium treatment induced an increase in Wnt/β-catenin pathway transcripts (HIG2, Bcl-xL, Cyclin D1 and c-myc) in the hippocampus and cerebral cortex of adult mice (De-Paula, V.J.,). Low-dose of lithium significantly improve spatial memory and ameliorate cognitive dysfunction and pathological alterations of Alzheimer's disease transgenic mice (APP/PS1) (Liu M, Qian T, Zhou W, Tao X, Sang S, Zhao L., 2020) and stimulates hippocampal glucose metabolism (Gherardelli C, Cisternas P, Inestrosa NC.).”
- The authors should improve the presentation of their data, I would suggest to use a program as Graphpad or R to report the graph (and they should insert a table with raw data of qPCR).
Thanks for the suggestion. The two graphs were redone in GraphPad Prism and the raw data of both graphs was inserted as an annex.
- If possible the authors should insert an image of cell (neuron) in culture at several concentration of lithium
Thanks for the suggestion, but it will not be possible to insert the cell images.
Reviewer 3 Report
This manuscript entitled "Lithium prevents telomere shortening in cortical neurons in amyloid beta-induced toxicity" discusses the neuroprotective effect of lithium in amyloid beta-induced toxicity in vitro by protecting telomere shortening. The article is not very original because the same group published in 2018 an article on a triple transgenic mouse model of AD where they described the effect of different doses of lithium on telomere maintenance in this model.
On the other hand, the results presented are weak and there is no information on the number of replicas, but it seems that not too many replicas have been used. It would be useful to include this information in the figure captions.
I have identified some errors:
- In the results part, you indicate that treatment with lithium is associated with greater viability of neurons, but statistical significance was only achieved with the medium dose, however, in the graph, there are asterisks in medium and high doses.
- In figure 2, despite the fact that you indicate that the Aß-peptide drastically decreases the length of telomeres, no asterisk appears on the graph. Is there any difference between control and treatment with Aß-42?
In addition, the publication instructions for authors indicate that a project report should include a research strategy and recommendations for future direction, but this manuscript lacks information on these topics. You should include a paragraph with information about new experiments to try to explain how lithium protects telomere shortening in this in vitro model.
Author Response
Reviewer 3
This manuscript entitled "Lithium prevents telomere shortening in cortical neurons in amyloid beta-induced toxicity" discusses the neuroprotective effect of lithium in amyloid beta-induced toxicity in vitro by protecting telomere shortening. The article is not very original because the same group published in 2018 an article on a triple transgenic mouse model of AD where they described the effect of different doses of lithium on telomere maintenance in this model.
We appreciate the suggestions and review of the work. In 2018, we published a study on brain tissue from a transgenic animal. We would like to point out that the present manuscript has the differential of presenting a primary culture of neurons, includes only neurons, so the work with brain tissue includes neurons and glial cells. Given this, I fear the direct action of lithium on the telomere length of neurons.
On the other hand, the results presented are weak and there is no information on the number of replicas, but it seems that not too many replicas have been used. It would be useful to include this information in the figure captions.
Thank you for the information and the number of the study (N=4) has been added to the figures.
I have identified some errors:
- In the results part, you indicate that treatment with lithium is associated with greater viability of neurons, but statistical significance was only achieved with the medium dose, however, in the graph, there are asterisks in medium and high doses.
Thanks for the comment. The MTT graph has been redone. In the graph we have the samples represented with * in relation to the control. The # is significance with respect to beta amyloid toxicity, where all treatments were significant.
- In figure 2, despite the fact that you indicate that the Aß-peptide drastically decreases the length of telomeres, no asterisk appears on the graph. Is there any difference between control and treatment with Aß-42?
The significance between control and Ab was added in the graph.
In addition, the publication instructions for authors indicate that a project report should include a research strategy and recommendations for future direction, but this manuscript lacks information on these topics. You should include a paragraph with information about new experiments to try to explain how lithium protects telomere shortening in this in vitro model.
Thanks for taking care to read the text. The following paragraph was added on page 7:
Future directions: new experiments in neurons from humans with Alzheimer's disease will be used in ongoing studies. With this we aim to explain how lithium protected the shortening of human telomeres.
Round 2
Reviewer 2 Report
The authors improved the quality of the manuscript, I would propose the acceptance in present form.